# Lipoic/Capsaicin-Related Amides: Synthesis and Biological Characterization of New TRPV1 Agonists Endowed with Protective Properties against Oxidative Stress

**DOI:** 10.3390/ijms232113580

**Published:** 2022-11-05

**Authors:** Antonella Brizzi, Samuele Maramai, Francesca Aiello, Maria Camilla Baratto, Federico Corelli, Claudia Mugnaini, Marco Paolino, Francesco Scorzelli, Carlo Aldinucci, Luciano De Petrocellis, Cinzia Signorini, Federica Pessina

**Affiliations:** 1Department of Biotechnology, Chemistry and Pharmacy, DoE 2018–2022, University of Siena, Via A. Moro 2, 53100 Siena, Italy; 2Department of Pharmacy, Health and Nutritional Sciences, University of Calabria, 87036 Arcavacata di Rende, Italy; 3Recipharm (Edmond Pharma), Strada Statale dei Giovi 131, 20037 Milano, Italy; 4Department of Molecular and Developmental Medicine, University of Siena, Via A. Moro 2, 53100 Siena, Italy; 5Endocannabinoid Research Group, Istituto di Chimica Biomolecolare, Consiglio Nazionale delle Ricerche, Via Campi Flegrei 34, 80078 Pozzuoli, Italy

**Keywords:** nutrient, dietary bioactive compound, lipoic acid, capsaicin, TRPV1, antioxidant, hypoxic-injury prevention, radical-scavenger

## Abstract

α-Lipoic acid is a sulfur-containing nutrient endowed with pleiotropic actions and a safe biological profile selected to replace the unsaturated alkyl acid of capsaicin with the aim of obtaining lipoic amides potentially active as a TRPV1 ligand and with significant antioxidant properties. Thus, nine compounds were obtained in good yields following a simple synthetic procedure and tested for their functional TRPV1 activity and radical-scavenger activity. The safe biological profile together with the protective effect against hypoxia damage as well as the in vitro antioxidant properties were also evaluated. Although less potent than capsaicin, almost all lipoic amides were found to be TRPV1 agonists and, specifically, compound **4**, the lipoic analogue of capsaicin, proved to be the best ligand in terms of efficacy and potency. EPR experiments and in vitro biological assays suggested the potential protective role against oxidative stress of the tested compounds and their safe biological profile. Compounds **4**, **5** and **9** significantly ameliorated the mitochondrial membrane potential caused by hypoxia condition and decreased F2-isoprostanes, known markers of oxidative stress. Thus, the experimental results encourage further investigation of the therapeutic potential of these lipoic amides.

## 1. Introduction

In the last decades, biologically active food constituents aroused research interest due to their ability to produce a plethora of pharmacological effects through the interaction with one or more biochemical pathways and the modulation of several metabolic processes. Importantly, both nutrients and non-nutritive compounds can contribute positively to the preservation of the health status or support, as adjuvant agents, established drug therapies, counteracting their side effects, and ameliorating the quality of life [1]. In this contest, there are several examples of non-nutritional uses of nutrients which produce therapeutically useful effects, such as the administration of folic acid in pregnant women or tryptophan supplementation to enhance serotonin synthesis and thereby promote sleep [2]. In this scenario, sulfur-containing nutrients emerged for their protective role against pathologies related to oxidative stress and inflammation, such as diabetes, cancer and cardiovascular and neurodegenerative diseases [3]. Within this class of compounds, α-lipoic acid (ALA, Figure 1), a small endogenous chiral disulfide endowed with pleiotropic actions and a safe biological profile, has gained attention for its chemical properties and therapeutic potential [4,5,6]. Once considered a vitamin, R-LA, the biologically active isomer, is synthesized in the liver and other tissues or introduced into the diet in the form of lipoyllysine. However, for therapeutic purposes, the racemic mixture is administered. The structure of this nutrient contains two oxidized (LA) or reduced (dihydrolipoic acid, DHLA) thiol groups that form a potent redox couple. These chemical features are responsible of its impressive array of antioxidant activities, i.e., as a scavenger of free radicals, a toxic metal chelator and a regenerator of other endogenous antioxidant molecules such as glutathione and vitamins C and E. Furthermore, both the oxidized and reduced forms show amphipathic properties being widely distributed in cytosol, cellular membranes, and the extracellular compartment [7,8].

Among the non-nutritive food compounds, capsaicin (Figure 1), the bioactive pungent ingredient of the hot chili pepper, represents a great example of a naturally occurring substance endowed with many beneficial biological properties including analgesic, anti-inflammatory, and antioxidant activities [9]. Many of its effects are due to the interaction with the transient potential vanilloid type-1 (TRPV1) channel, a polymodal nociceptor, which is first activated and then desensitized. Currently used only topically in the treatment of pain and skin disorders, the main drawbacks for therapeutic purposes are represented by the low bioavailability, the intense burning sensation in the mucosa and the cardiovascular side effects [10]. Structurally, capsaicin, or *trans*-8-methyl-*N*-vanillyl-6-nonenamide, is characterized by a lipophilic acid tail joined through an amide bond to a vanillyl head, and has features conferring low water solubility and high volatility.

Searching for appropriate substitutes of the unsaturated alkyl acid of capsaicin, ALA was selected as a versatile molecule [11] that can easily be conjugated with opportune amines to produce lipoic amides potentially active as TRPV1 ligands and with significant antioxidant properties. In fact, in a previous work [12] we identified a new class of TRPV1 agonists replacing the nonenoyl portion of capsaicin by a moderately flexible aromatic backbone and maintaining an aromatic amide head suitably decorated with H-bond donor/acceptor substituents. Thus, with the aim of merging the specific properties of both natural compounds, a set of α-lipoic acid amides were synthetized using several aromatic amines bearing groups with different electronic and steric properties. Specifically, methylcatechol and catechol systems or the hydroxyl group alone in the *para* position were chosen for conferring antioxidant activity and a conceivable ability to interact with the TRPV1 receptor, while halogens were evaluated as a valid alternative (Figure 1). All final compounds were tested for their functional TRPV1 activity, and their structure-activity relationship analysis is discussed.

Moreover, the safe biological profile together with the protective effect against hypoxia damage as well as the in vitro antioxidant properties and scavenger profile of the opportune derivatives were also evaluated.

## 2. Results

### 2.1. Chemistry

ALA was simply converted into corresponding amides using the selected amines in the presence of either *N*-ethyl-*N*’-(3-dimethylaminopropyl)carbodiimide (EDCI) or O-(benzotriazol-1-yl)-*N*,*N*,*N*′,*N*′-tetramethyluronium hexafluorophosphate (HBTU) as coupling reagents in the appropriate conditions. In greater detail, free amines (4-aminophenol, 4-fluoroaniline, 2,4-difluoroaniline, 3,4-dichlorobenzylamine, 4-methoxybenzylamine, 4-hydroxybenzylamine, 4-fluorobenzylamine) were coupled with ALA in the presence of EDCI and 1-hydroxybenzotriazole (HOBt) using dry dichloromethane (DCM) as the solvent (conditions A); conversely, amine hydrochlorides (vanillylamine hydrochloride and dopamine hydrochloride) were reacted with ALA in dry *N*,*N*-dimethylformamide (DMF) using HBTU, HOBt and *N*,*N*-diisopropylethylamine (DIPEA) to improve solubility and increase yields (conditions B, Figure 1).

Within this set of amides, derivatives **4** and **7** are novel compounds, four derivatives (**3**, **5**, **6** and **8**) are found to be commercial substances, and finally, three derivatives (**1**, **2** and **9**) are known compounds with literature documents available [13,14,15,16]., The identity of all synthetized lipoic amides was unambiguously assessed by spectroscopic data which were in agreement with the proposed chemical structure and with those already present in the literature.

### 2.2. Functional TRPV1 Assay

Final amides **1–9**, along with capsaicin, were tested for their TRPV1 interaction profile through a functional assay that uses human embryonic kidney (HEK293) cells stably overexpressing recombinant human TRPV1 [17]. The increase in intracellular calcium concentration was employed as a measure of both activation (EC_50_) and desensitization (IC_50_ determined against the effect of capsaicin after 5 min of preincubation of each compound) effects, and the results are listed in Table 1. Interestingly, all lipoic amides, with the exception of derivative **3**, were found to be able to interact with the TRPV1 channel, behaving as agonists with varying potency.

### 2.3. Scavenger Activity through the DPPH Assay Using EPR Spectroscopy

Electron Paramagnetic Resonance (EPR) spectroscopy allows for the acquisition of the scavenger activity of a compound endowed with potential antioxidant activity. This is possible through the study of the reduction of the signal intensity generated by a free radical before and after the addition of the test compound and is acquired by the EPR spectrum.

Since compounds **1**, **4–7** and **9** possess the ideal structural features for free radical-scavenger activity, the EPR technique was applied to determine their scavenger activity towards the free DPPH (2,2-diphenyl-1-picrylhydrazyl) radical.

In Figure 2, the recorded EPR spectra of the DPPH radical signals for the two series of synthesized compounds, i.e., **1**, **4–5** (**a**) and **6–7** and **9** (**b**), are reported. The calculated values of the scavenger ratio percentage R in decreasing order were: 99.0% for compound **9**, 69.9% for compound **1**, 39.1% for compound **4**, 26.0% for compound **7**, 9.5% for compound **5** and 4.0% for compound **6**. Under these same conditions the R value for capsaicin was 93% (Appendix A).

### 2.4. In Vitro Biological Assays

The cytotoxicity of the compounds was evaluated in human immortal keratinocytes (HaCat) cells. To quantitatively measure cellular viability, the fluorescein diacetate method was used. Results showed that all the compounds (range 0.1–10 µM) were devoid of cytotoxicity, as cell viability was mostly unchanged 48 h after treatment, demonstrating their safe profile even after a prolonged drug exposure (Appendix A).

Compounds **1**, **4**–**5**, **7** and **9** and capsaicin were also tested for their protective effect against hypoxic damage. However, compound **6** was not tested because it showed the lowest R value in the DPPH assay as well as low TRPV1 activity. Hypoxia causes a decrease in ATP synthesis and increased reactive oxygen species (ROS) formation, giving rise to oxidative stress that usually leads to mitochondrial membrane potential (MMP) modification, thus inducing cell death. HaCat cells were thus subjected to a hypoxic-mediated injury (2% O_2_, 5% CO_2_ and 93% N_2_ for 48 h) in the presence or absence of capsaicin or compounds (**1**, **4**–**5**, **7**, **9**), concentrations ranging from 0.1 to 10 µM, and at the end of the treatment, cell viability and MMP modification were assessed.

Results showed that in the hypoxia condition there was a decrease in the viability (about 30% cell death) and a significant increase of MMP (about 50%). Unfortunately, not all the compounds were able to protect the cells from death. Interestingly, while only capsaicin and compound **9** were able to significantly increase cell viability (Figure 3), the MMP returned to control values in the presence of four of the five tested compounds. In fact, compound **4** and **5** significantly reduced MMP at 1 and 3 µM, while the higher concentration of 10 µM was less effective; on the contrary, compounds **1** and **9** showed a protective effect at the higher concentrations used (Figure 4).

The protective effects exerted by the compounds against hypoxia damage led us to look at markers of oxidative stress such as F2-isoprostanes (F2-IsoPs), thus allowing a better understanding of the results obtained. As represented in Figure 5, in the hypoxia condition the levels of F2-IsoPs significantly increased; interestingly, all the tested compounds were able to significantly decrease F2-IsoPs, bringing them back to control values.

## 3. Discussion

The initial hypothesis of replacing the naturally occurring fatty acid with lipoic acid in capsaicin led to the synthesis of a set of nine lipoic amides that were evaluated for the first time for their ability to interact with the TRPV1 channel. The experimental results obtained from the functional TRPV1 assay (Table 1) showed that almost all synthesized derivatives were able to bind the receptor behaving as agonists with efficacy and potency values ranging from low to good. Although less potent than capsaicin, the best ligand in terms of efficacy (71.1%, see Table 1) and potency (0.13 µM) was derivative 4, characterized by the vanillyl moiety in the amide head as in the natural agonist. All lipoic amides showing high to moderate efficacy, i.e., derivatives **1**–**2**, **4**–**5** and **8**, are characterized by the *para* position substituted either with a hydroxyl group or with a halogen whichresulted well tolerated. Specifically, compounds **2** and **5** also maintained acceptable potency (1.87 and 1.7 µM, respectively). On the contrary, methylation of the hydroxyl in the *para* position, as in derivative 6, which present a methoxy substituent, caused a drastic drop in the potency value (20.5 µM), while maintaining an efficacy of about 53%. Even more detrimental for the ligand-receptor interaction, it turned out to be the introduction of a substituent in position 2 of the aromatic nucleus, providing the only completely inactive derivative 3. Concerning the distance of the aromatic nucleus from the amide functionality, the benzyl moiety, having a methylene spacer (*n* = 1), was confirmed to confer the best interaction features (compounds **4** and **5**). While the shortening of the spacer (compounds **1** and **2**, *n* = 0) was well tolerated, its lengthening led to a derivative characterized by poor efficacy and lower potency (compound **9**, *n* = 2). Afterwards, the lipoic amides **1**, **4–7** and **9** possessing the crucial structural arrangements potentially imparting a high antioxidant activity, as well as capsaicin, were also evaluated for their ability to scavenge the free DPPH radical by EPR analysis. Calculated R values indicated that in this set of compounds the catecholic system conferred the best radical-scavenger activity, lowering the signal intensity by as much as 99% (compound **9**, Figure 2b). Indeed, compound **9** showed better antiradical activity than capsaicin itself (R = 93%, Appendix A). Among the 4′-substituted derivatives, the ability to scavenge the free DPPH radical decreased progressively from compound **1**, which still retained a rather high R value (about 70%), to compounds **7** and **6** (R = 26.0% and 4.0%, respectively); this trend in the R values highlighted that the free hydroxyl (compounds **1** and **7**) is a better antioxidant substituent than the methoxy group (compound **6**) as well as the 4-hydroxyphenyl moiety (compound **7**) compared to the 4-hydroxybenzyl moiety (compound **6**). Despite the fact that compound **4** retained an acceptable antiradical power (R value about 39%), the methylation of one of the two hydroxyls in the cathecol system or their replacement with two chlorine atoms (compound **5**) significantly decreased the antioxidant activity. Moreover, capsaicin was found to have a significantly higher scavenger activity than its lipoic analogue 4 (93% and 39%, respectively), suggesting that the nonenoyl moiety exhibits better anti-scavenger properties than lipoyl residue when coupled with the vanillylamine.

HaCat cells are a good model to test the in vitro biological effect of capsaicin and our synthesized derivatives as they express TRPV1 receptors with which the compounds could interact. The obtained cell viability results showed that all the tested compounds, i.e., **1**, **4–5**, **7** and **9**, were devoid of cytotoxicity, thus suggesting their safe biological profile. Moreover, as compounds **1**, **4–5**, **7** and **9** are endowed with antiradical activity as shown by EPR results, their potential protection against hypoxic damage was also investigated. Hypoxia causes a decrease in ATP synthesis and increases ROS formation, giving rise to oxidative stress that leads to cell death. HaCat cells subjected to a hypoxic-mediated injury showed a moderate decrease in cell viability and a modification of MMP. Normally, the mitochondrial membrane is polarized, and the membrane potential is maintained by ATP and proton gradients, while ROS are compartmentalized within the mitochondria. After hypoxia, ROS are released into the cytoplasm causing a variety of signal molecules. Moreover, along with the modification of MMP, there is also an increase of markers of oxidative stress such as F2-IsoPs, which might have many effects on the biophysical processes of cell membranes since free radicals produced by the arachidonic acid metabolism contribute to the irreversible depolarization produced by ischemia [18].

All of the compounds tested in our study ameliorated the mitochondrial membrane potential caused by the hypoxia condition; compounds **4**, **5** and **9** seemed the most active. Interestingly, all the compounds were able to significantly decrease F2-IsoPs, bringing them back to control values and thus suggesting their protective role against oxidative stress.

## 4. Materials and Methods

### 4.1. Chemistry

All starting materials, including solvents, were purchased from common commercial suppliers and were used without further purification. Organic solutions, after being dried over anhydrous sodium sulphate, were concentrated with a Büchi rotary evaporator (R-110, Milan, Italy) equipped with a vacuum pump (KNF N 820 FT 18). Final amides were purified by a Biotage flash chromatography system with columns 12.25 mm, packed with KP-Sil, 60A, 32–63 μm and checked for purity by TLC on Merck 60 F254 silica plates (Milan, Italy). Melting points were determined on a Kofler apparatus (K) and are uncorrected. The identity of compounds was unambiguously assessed by nuclear magnetic resonance and mass spectroscopy. ^1^H-NMR and ^13^C-NMR spectra were recorded in the indicated solvent at 25 °C, both on a Bruker Advance (Milan, Italy) operating at 600 MHz and a Bruker Advance DPX400 (Milan, Italy), and chemical shifts (δ) were expressed in ppm and the coupling constants (J) in Hz. Mass spectra (MS) were acquired either in the positive or the negative mode (mass range *m/z* of 150–1500) using an Agilent 1100 LC/MSD VL system (Milan, Italy) with a 0.4 mL/min flow rate of a binary eluent mixture (95/5 methanol/water) and supplied with a UV detector (254 nm). Compound purity was assessed by elemental analysis on a Perkin-Elmer PE 2004 analyzer (Milan, Italy), and the data for C, H, N are within ± 0.4% of the theoretical values.

#### 4.1.1. General Procedure for the Synthesis of Lipoic Amides **1–9**

Conditions A. A solution of the opportune free amine (1.5 eq., 4-aminophenol, 4-fluoroaniline, 2,4-difluoroaniline, 3,4-dichlorobenzylamine, 4-methoxybenzylamine, 4-hydroxybenzylamine, 4-fluorobenzylamine) in dry dichloromethane (DCM, 10 mL) was added to a solution of ALA (1.0 eq.) in the same solvent (20 mL) kept at room temperature (rt) and under a positive dry nitrogen atmosphere; subsequently, HOBt (1.2 eq.) and EDCI (1.5 eq.) were added, leaving the reaction under stirring overnight. The reaction mixture was then diluted with DCM. The organic layer was washed twice with a NH_4_Cl saturated solution and once with brine, dried over sodium sulfate, filtered, and evaporated to dryness. The collected raw material was purified by flash chromatography in the indicated solvent furnishing the final amides (compounds **1–3** and **5–8**).Conditions B. To a solution of ALA (1.0 eq.) in anhydrous DMF (10 mL), kept under a positive pressure of dry nitrogen, were added in the order HBTU (2.0 eq.), HOBt (1.0 eq.), DIPEA (1.5 eq.) and the amine hydrochloride (1.2 eq., vanillylamine hydrochloride or dopamine hydrochloride) and the reaction was stirred for 40 min at rt; DIPEA (1.5 eq.) was then added again, leaving the reaction to stand overnight. Next, the reaction mixture was diluted with chloroform and the organic phase was washed several times (x4) with a NH_4_Cl saturated solution, and finally once with brine. After drying and filtration, the solvent was concentrated to give a crude residue which was purified by flash chromatography using a mixture chloroform/methanol (50/2) as eluent (compounds **4** and **9**).

#### 4.1.2. 5-(1,2-Dithiolan-3-yl)-*N*-(4-Hydroxyphenyl)Pentanamide (1)

Eluent, CHCl_3_/CH_3_OH 48/2. Yield, 80.0%. Yellow solid, mp 105–107 °C (K). ^1^H-NMR (400 MHz, acetone-*d_6_*): δ 8.83 (s, 1H), 8.04 (s, 1H), 7.43 (d, 2H, *J* = 8.8 Hz), 6.72 (d, 2H, *J* = 8.8 Hz), 3.62–3.55 (m, 1H), 3.20–3.14 (m, 1H), 3.12–3.06 (m, 1H), 2.49–2.41 (m, 1H), 2.30 (t, 2H, *J* = 7.3 Hz), 1.92–1.84 (m, 1H), 1.77–1.58 (mm, 4H), 1.50–1.42 (m, 2H). ^13^C-NMR (100 MHz, acetone-*d_6_*): δ 170.3 (C=O), 153.4 (Cq), 131.8 (Cq), 120.9 (x2, CH-Ar), 115.0 (x2, CH-Ar), 56.4 (CH), 40.0 (CH_2_), 38.2 (CH_2_), 36.5 (CH_2_), 34.6 (CH_2_), 28.7 (CH_2_, under solvent signals), 25.2 (CH_2_). ESI-MS *m/z*: 298 [M+H]^+^ (25), 320 [M+Na]^+^ (100). Elemental analysis for C_14_H_19_NO_2_S_2_ found C, 56.42; H, 6.46; N, 4.70; calculated C, 56.54; H, 6.44; N, 4.71.

#### 4.1.3. 5-(1,2-Dithiolan-3-yl)-*N*-(4-Fluorophenyl)Pentanamide (2)

Eluent, CHCl_3_. Yield, 85.0%. Waxy yellow solid, mp 82–84 °C (K). ^1^H-NMR (600 MHz, chloroform-*d*): δ 7.97 (s, 1H), 7.45 (dd, 2H, *J* = 8.8, 4.8 Hz), 6.96 (t, 2H, *J* = 8.6 Hz), 3.55–3.51 (m, 1H), 3.17–3.13 (m, 1H), 3.11–3.07 (m, 1H), 2.45–2.40 (m, 1H), 2.33 (t, 2H, *J* = 7.4 Hz), 1.90–1.84 (m, 1H), 1.77–1.63 (mm, 4H), 1.52–1.42 (m, 2H). ^13^C-NMR (150 MHz, chloroform-*d*): δ 171.6 (C=O), 159.4 (d, Cq-F, *J* = Hz), 134.0 (Cq), 122.1 (x2, CH-Ar), 115.5 (x2, CH-Ar), 56.4 (CH), 40.3 (CH_2_), 38.5 (CH_2_), 37.1 (CH_2_), 34.6 (CH_2_), 28.9 (CH_2_), 25.3 (CH_2_). ESI-MS *m/z*: 322 [M+Na]^+^ (100). Elemental analysis for C_14_H_18_FNOS_2_ found C, 56.04; H, 6.08; N, 4.69; calculated C, 56.16; H, 6.06; N, 4.68.

#### 4.1.4. *N*-(2,4-Difluorophenyl)-5-(1,2-Dithiolan-3-yl)Pentanamide (3)

Eluent, CHCl_3_. Yield, 60.0%. Waxy yellow solid, mp 79–81 °C (K). ^1^H-NMR (600 MHz, chloroform-*d*): δ 8.20 (dd, 1H, *J* = 14.8, 8.5 Hz), 7.33 (br s, 1H), 6.86–6.83 (m, 2H), 3.60–3.55 (m, 1H), 3.19–3.17 (m, 1H), 3.13–3.09 (m, 1H), 2.48–2.43 (m, 1H), 2.41 (t, 2H, *J* = 7.1 Hz), 1.94–1.88 (m, 1H), 1.80–1.66 (mm, 4H), 1.57–1.49 (m, 2H). ^13^C-NMR (150 MHz, chloroform-*d*): δ 171.1 (C=O), 158.6 (d, Cq-F, *J* = 246.6 Hz), 152.5 (d, Cq-F, *J* = 246.1 Hz), 123.1 (CH-Ar), 122.6 (Cq), 111.2 (CH-Ar), 103.5 (CH-Ar), 56.4 (CH), 40.3 (CH_2_), 38.5 (CH_2_), 37.2 (CH_2_), 34.7 (CH_2_), 28.9 (CH_2_), 25.2 (CH_2_). ESI-MS *m/z*: 340 [M+Na]^+^ (100). Elemental analysis for C_14_H_17_F_2_NOS_2_ found C, 53.06; H, 5.42; N, 4.40; calculated C, 52.98; H, 5.40; N, 4.41.

#### 4.1.5. 5-(1,2-Dithiolan-3-yl)-*N*-(4-Hydroxy-3-Methoxybenzyl)Pentanamide (4)

Eluent, CHCl_3_/CH_3_OH 50/2. Yield, 75.0%. Pasty yellow oil. ^1^H-NMR (400 MHz, chloroform-*d*): δ 6.79 (d, 1H, *J* = 8.0 Hz), 6.74 (s, 1H), 6.69 (d, 1H, *J* = 8.0 Hz), 6.01 (br s, 1H), 4.28 (d, 2H, *J* = 5.5 Hz), 3.80 (s, 3H), 3.52–3.45 (m, 1H), 3.14–3.08 (m, 1H), 3.07–3.01 (m, 1H), 2.42–2.34 (m, 1H), 2.16 (t, 2H, *J* = 7.4 Hz), 1.87–1.79 (m, 1H), 1.69–1.55 (mm, 4H), 1.46–1.33 (m, 2H). ^13^C-NMR (100 MHz, chloroform-*d*): δ 172.8 (C=O), 146.8 (Cq), 145.2 (Cq), 130.2 (Cq), 120.8 (CH-Ar), 114.5 (CH-Ar), 110.9 (CH-Ar), 56.4 (CH), 56.0 (OCH_3_), 43.6 (CH_2_-benz), 40.2 (CH_2_), 38.5 (CH_2_), 36.4 (CH_2_), 34.6 (CH_2_), 28.9 (CH_2_), 25.4 (CH_2_). ESI-MS *m/z*: 342 [M+H]^+^ (100), 364 [M+Na]^+^ (45). Elemental analysis for C_16_H_23_NO_3_S_2_ found C, 56.16; H, 6.81; N, 4.11; C, 56.28; H, 6.79; N, 4.10.

#### 4.1.6. *N*-(3,4-Dichlorobenzyl)-5-(1,2-Dithiolan-3-yl)Pentanamide (5)

Eluent, CHCl_3_/CH_3_OH 50/1. Yield, 86.0%. Yellow solid, mp 84–86 °C (K). ^1^H-NMR (600 MHz, chloroform-*d*): δ 7.38 (d, 1H, *J* = 8.2 Hz), 7.35 (d, 1H, *J* = 1.6 Hz), 7.11 (dd, 1H, *J* = 8.2, 1.7 Hz), 6.03 (br s, 1H), 4.37 (d, 2H, *J* = 5.8 Hz), 3.58–3.53 (m, 1H), 3.19–3.15 (m, 1H), 3.13–3.09 (m, 1H), 2.47–2.42 (m, 1H), 2.24 (t, 2H, *J* = 7.4 Hz), 1.92–1.87 (m, 1H), 1.73–1.64 (mm, 4H), 1.49–1.42 (m, 2H). ^13^C-NMR (150 MHz, chloroform-*d*): δ 172.8 (C=O), 138.8 (Cq), 132.7 (Cq), 131.5 (Cq), 130.6 (CH-Ar), 129.6 (CH-Ar), 127.1 (CH-Ar), 56.4 (CH), 42.4 (CH_2_-benz), 40.2 (CH_2_), 38.5 (CH_2_), 36.3 (CH_2_), 34.6 (CH_2_), 28.9 (CH_2_), 25.3 (CH_2_). ESI-MS *m/z*: 365 [M+H]^+^ (30), 387 [M+Na]^+^ (100). Elemental analysis for C_15_H_19_Cl_2_NOS_2_ found C, 49.57; H, 5.28; N, 3.84; calculated C, 49.45; H, 5.26; N, 3.84.

#### 4.1.7. 5-(1,2-Dithiolan-3-yl)-*N*-(4-Methoxybenzyl)Pentanamide (6)

Eluent, CHCl_3_/CH_3_OH 50/2. Yield, 92.0%. Pale yellow solid, mp 69–71 °C (K). ^1^H-NMR (600 MHz, chloroform-*d*): δ 7.16 (d, 2H, *J* = 8.4 Hz), 6.82 (d, 2H, *J* = 8.4 Hz), 6.16 (br s, 1H), 4.31 (d, 2H, *J* = 5.1 Hz), 3.76 (s, 3H), 3.54–3.50 (m, 1H), 3.16–3.12 (m, 1H), 3.10–3.05 (m, 1H), 2.44–2.39 (m, 1H), 2.18 (t, 2H, *J* = 7.5 Hz), 1.89–1.83 (m, 1H), 1.68–1.60 (mm, 4H), 1.46–1.39 (m, 2H). ^13^C-NMR (150 MHz chloroform-*d*): δ 172.8 (C=O), 158.9 (Cq), 130.6 (Cq), 129.2 (CH-Ar), 129.0 (CH-Ar), 114.1 (CH-Ar), 114.0 (CH-Ar), 56.4 (CH), 55.3 (OCH_3_), 42.9 (CH_2_-benz), 38.5 (x2, CH_2_), 36.3 (CH_2_), 34.6 (CH_2_), 28.9 (CH_2_), 25.4 (CH_2_). ESI-MS *m/z*: 326 [M+H]^+^ (20), 348 [M+Na]^+^ (100). Elemental analysis for C_16_H_23_NO_2_S_2_ found C, 59.10; H, 7.14; N, 4.29; calculated C, 59.04; H, 7.12; N, 4.30.

#### 4.1.8. 5-(1,2-Dithiolan-3-yl)-*N*-(4-Hydroxybenzyl)Pentanamide (7)

Eluent, EtOAc. Yield, 70.0%. Pale yellow oil. ^1^H-NMR (600 MHz, methanol-*d_4_*): δ 7.11 (d, 2H, *J* = 8.4 Hz), 6.73 (d, 2H, *J* = 8.4 Hz), 4.25 (s, 2H), 3.57–3.52 (m, 1H), 3.18–3.14 (m, 1H), 3.11–3.07 (m, 1H), 2.45–2.40 (m, 1H), 2.22 (t, 2H, *J* = 7.3 Hz), 1.88–1.83 (m, 1H), 1.73–1.60 (mm, 4H), 1.50–1.38 (m, 2H). ^13^C-NMR (150 MHz, methanol-*d_4_*): δ 174.3 (C=O), 156.4 (Cq), 129.5 (Cq), 128.7 (x2, CH-Ar), 114.8 (x2, CH-Ar), 56.2 (CH), 42.3 (CH_2_-benz), 39.9 (CH_2_), 38.0 (CH_2_), 35.4 (CH_2_), 34.3 (CH_2_), 28.4 (CH_2_), 25.4 (CH_2_). ESI-MS *m/z*: 334 [M+Na]^+^ (100). Elemental analysis for C_15_H_21_NO_2_S_2_ found C, 57.73; H, 6.82; N, 4.71; calculated C, 57.85; H, 6.80; N, 4.50.

#### 4.1.9. 5-(1,2-Dithiolan-3-yl)-*N*-(4-Fluorobenzyl)Pentanamide (8)

Eluent, CHCl_3_. Yield, 75.0%. Pasty yellow solid, mp 70–72 °C (K). ^1^H-NMR (600 MHz, chloroform-*d*): δ 7.21 (dd, 2H, *J* = 8.2, 5.5 Hz), 6.97 (t, 2H, *J* = 8.6 Hz), 6.20 (br s, 1H), 4.35 (d, 2H, *J* = 5.7 Hz), 3.55–3.50 (m, 1H), 3.16–3.12 (m, 1H), 3.10–3.06 (m, 1H), 2.45–2.39 (m, 1H), 2.19 (t, 2H, *J* = 7.4 Hz), 1.89–1.84 (m, 1H), 1.70–1.61 (mm, 4H), 1.48–1.38 (m, 2H). ^13^C-NMR (150 MHz, chloroform-*d*): δ 172.8 (C=O), 162.1 (d, Cq-F, *J* = 245.6 Hz), 134.3 (Cq), 129.5 (CH-Ar), 129.3 (CH-Ar), 115.7 (CH-Ar), 115.3 (CH-Ar), 56.4 (CH), 42.8 (CH_2_-benz), 40.2 (CH_2_), 38.5 (CH_2_), 36.3 (CH_2_), 34.6 (CH_2_), 28.9 (CH_2_), 25.4 (CH_2_). ESI-MS *m/z*: 314 [M+H]^+^ (100), 336 [M+Na]^+^ (40), 352 [M+K]^+^ (18). Elemental analysis for C_15_H_20_FNOS_2_ found C, 57.60; H, 6.45; N, 4.46; calculated C, 57.48; H, 6.43; N, 4.47.

#### 4.1.10. *N*-(3,4-Dihydroxyphenethyl)-5-(1,2-Dithiolan-3-yl)Pentanamide (9)

Eluent, CHCl_3_/CH_3_OH 48/2. Yield, 55.0%. Chewy yellow oil. ^1^H-NMR (600 MHz, methanol-*d_4_*): δ 6.68 (d, 1H, *J* = 8.0 Hz), 6.65 (d, 1H, *J* = 2.3 Hz), 6.53 (dd, 1H, *J* = 7.9, 2.2 Hz), 3.57–3.52 (m, 1H), 3.34 (t, 2H, *J* = 7.3 Hz), 3.18–3.15 (m, 1H), 3.11–3.07 (m, 1H), 2.64 (t, 2H, *J* = 7.3 Hz), 2.47–2.42 (m, 1H), 2.15 (t, 2H, *J* = 7.4 Hz), 1.89–1.84 (m, 1H), 1.71–1.65 (m, 1H), 1.64–1.55 (mm, 3H), 1.46–1.34 (m, 2H). ^13^C-NMR (150 MHz, methanol-*d_4_*): δ 174.6 (C=O), 144.9 (Cq), 143.4 (Cq), 130.6 (Cq), 119.7 (CH-Ar), 115.5 (CH-Ar), 115.0 (CH-Ar), 56.1 (CH), 40.7 (CH_2_), 39.9 (CH_2_), 38.0 (CH_2_), 35.5 (CH_2_), 34.5 (CH_2_), 34.4 (CH_2_), 28.4 (CH_2_), 25.4 (CH_2_). ESI-MS *m/z*: 364 [M+Na]^+^ (100), 380 [M+K]^+^ (15). Elemental analysis for C_16_H_23_NO_3_S_2_ found C, 56.14; H, 6.81; N, 4.11; calculated C, 56.28; H, 6.79; N, 4.10.

### 4.2. Radical-Scavenger Activity: DPPH Assay and EPR Analysis

X-band (9GHz) EPR measurements were performed to determine the scavenger activity of the samples **1**, **4–7** and **9**, and capsaicin towards the DPPH (2,2-diphenyl-1-picrylhydrazyl) radical. All solutions were prepared in acetonitrile with a final concentration of 0.1 mM of DPPH and 2.2 mM of the test compound. The spectra were recorded at room temperature using a Bruker E580 Elexsys Series and a Bruker cavity ER 4122 SHQE. The compounds, and capsaicin, were added to the radical solution and after an incubation time of 20 min the EPR spectrum was recorded. The signal of the DPPH before and after the addition of each compound was acquired and the relative area or double integral of the EPR spectra were calculated. The scavenger activity of the samples was calculated as follows:$$R=\frac{H_{0}-H_{x}}{H_{0}}\times 100$$
where *H*_0_ and *H_x_* are the double integral or area of the EPR signal in the absence or after the addition of the antioxidant compound, respectively, and the scavenger ratio (R) represents the percentage of scavenger activity of the samples. The reduction of the intensity of the signals is much more evident for those samples which have higher antioxidant activity.

### 4.3. In Vitro Biological Assays

#### 4.3.1. Cell Cultures and Drug Treatments

Human immortal keratinocytes (HaCaT) cell lines were cultured in standard conditions until 80% confluency, as previously described [19].

Compounds and capsaicin were prepared as a 10 mM stock solution in dimethyl sulfoxide (DMSO) and diluted to the required final concentrations with cell culture medium immediately before use. The final concentration of DMSO was kept below 0.1% (*v/v*). The appropriate controls, run in parallel to the experiments, indicated that the vehicle neither affected cell viability nor interfered with the assays.

#### 4.3.2. Hypoxia-Induced Injury

HaCat cells, treated with medium, with capsaicin or with compounds (0.1–10 µM), were placed in a humidified incubator (Sanyo, Osaka, Japan) with gas concentrations of 2% O_2_, 5% CO_2_ and 93% N_2_ for 48 h. Control cells were kept in a separate incubator (95% air: 5% CO_2_) for the same period. After hypoxia, cell viability, mitochondria membrane potential (MMP) and the content of F_2_-IsoPs were assessed.

#### 4.3.3. Cell Viability Assay

HaCaT cells were incubated with capsaicin or compounds for 48 h, and afterward cells viability was assessed with fluorescein diacetate and Hoechst 33342 (Sigma Chemical, Dorset, UK). Briefly, these substances were added to cells at final concentrations of 1 and 10 μg/mL for propidium iodide, fluorescein diacetate and Hoechst 33342, respectively. Fluorescence was examined with a Fluoroskan Ascent fluorimeter (ThermoLabsystems, Helsinki, Finland) at 485 nm excitation and 538 nm emission, and 355 nm excitation and 460 nm emission for fluorescein diacetate and Hoechst 33342, respectively.

#### 4.3.4. Determination of MMP

HaCat cells were loaded with JC-1 (tetraethylbenzimidazolylcarbocyanine iodide, 6 μM) for 12 min at 37 °C and then washed in phosphate-buffered saline (pH 7.4). The fluorescence was recorded with a Fluoroskan Ascent fluorimeter (ThermoLabsystems, Helsinki, Finland). Control cells were treated as above. During the measurement, cells were maintained at 37 °C and protected from light. The MPP was estimated as the ratio of JC-1 aggregates (red, λ_em_. = 590 nm) at λ_ex_. = 485 nm to JC-1 monomers (green, λ_em_. = 538 nm). All fluorescence measurements were corrected for the autofluorescence of cells not loaded with JC-1, which was constant throughout the experiments. In control experiments, no photobleaching was observed during fluorescence measurements.

#### 4.3.5. Determination of Total F2-Isoprostanes

F_2_-IsoPs, a series of prostaglandin F_2_-like compounds mainly generated by non-enzymatic free radical-initiated peroxidation of membrane arachidonic acid, are measured to evaluate lipid peroxidation. The determination of total F_2_-IsoPs was performed by gas chromatography/negative ion chemical ionization tandem mass (GC/NICI–MS/MS), as reported. Briefly, HaCat cells (treated as specified above) were lysed in the presence of 100 μM butylhydroxytoluene (BHT). Afterward, basic hydrolysis was carried out by incubation (45 °C for 45 min) with 1 N KOH; 1 N HCl, and tetradeuterated prostaglandin F_2α_ (PGF_2α_-d_4_) (500 pg) was added, and an extraction in the presence of ethyl acetate (10 mL) was performed. In the following purification procedures, the obtained total lipid extract was transferred to an NH_2_ cartridge, and procedures of conditioning, washing, and elution were performed. The collected eluates were evaporated under nitrogen at 40 °C, and two derivatization processes were carried out to form pentafluorobenzyl ester and trimethylsilyl ether groups [20]. GC/NICI–MS/MS analysis was used to measure the ion at *m/z* 299 produced from 15-F_2t_-IsoP, one of the most represented isomers of F_2_-IsoPs. The quantitation of total F_2_-IsoPs was determined by relating the measured amounts of 15-F_2t_-IsoP (Cayman Chemical, Item No. 16350) to the calibration curves constructed.

## 5. Conclusions

A set of lipoic amides were synthesized and evaluated for their functional activity at the TRPV1 channel. Almost all derivatives were found to interact with the TRPV1 receptor, behaving as agonists. In particular, compound **4**, chemically the lipoic analogue of capsaicin, proved to be the best ligand in terms of efficacy and potency. Interestingly, the assessment of the antiradical properties of the amides, which possess crucial antioxidant features, indicated that catecholic (compound **9**) and methylcatecholic (compound **4**) systems conferred the best radical-scavenger activity. Moreover, in vitro biological assays showed that all tested derivatives, of which compounds **4**, **5** and **9** seemed the most active, ameliorated the mitochondrial membrane potential caused by hypoxia condition and lowered F_2_-IsoPs levels to the control values, suggesting their protective role against oxidative stress. Overall, the results presented herein confirm that ALA is a versatile molecule and a suitable acyl donor to replace the nonenoyl residue of capsaicin, furnishing amides endowed with a safe biological profile and an interesting pharmacological activity which deserves to be investigated further.

## 6. Patents

Brizzi, A.; Aiello, F.; Corelli, F. Ligandi TRPV1. **2014**, Patent N° 0001424275 (FI 2014 A000096).

## Data Availability

Data is contained within the article or Appendix A.

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
