# Peer review of "Lipoic/Capsaicin-Related Amides: Synthesis and Biological Characterization of New TRPV1 Agonists Endowed with Protective Properties against Oxidative Stress"

_ijms, 2022, doi:10.3390/ijms232113580_

Round 1

Reviewer 1 Report

The manuscript “Lipoic/capsaicin-related amides. Synthesis and biological characterization of new TRPV1 agonists endowed with protective 3 properties against oxidative stress." submitted to International Journal for Molecular Sciences brings description nine lipoyl amides as potential TRPV1 agonists with protective role against oxidative stress and possible therapeutic potential. The manuscript is well written and easy to follow. However, I have several concerns listed in the comments.

General comments:

It is commendable that the authors have researched and presented this kind of research. My major comments that concern me are:

1. Why didn’t you conduct further experiments after DPPH assay on compound 6? Add good enough explanation in the text.

2.  You haven’t used positive controls in experiments, only vehicle? Why? In all  asseys there should always be positive control(s) as the control of the test itself.

Specific comments

1.            In Table 1 add “% “after TRPV1 Efficacy? - TRPV1 Efficacy (%)

2.            Text from line 136-148 is for Materials and Methods section, transfer it there.

3.            Move text from line 156-160 before Figure 2, since you are commenting Figure 2.

4.            Paraphrase “MMP returned to values like that”, line 180.

5.            Regardless of the statistics, I don't think you cannot state that something is protective given such small difference in Figure 3. Accordingly adjust in text.

6.            How do you explain that different concentrations are cytoprotective and modify MMP?

7.            In line 222 and 224  use "high" instead "good".

8.            In line 225, you stated that compound 8 is “good efficacy”, but you haven't tested it further. Either mitigate or reformulate.

9.            In line 255, unbold in vitro.

10.          Add viability assay in description “Supplementary Materials: The following are available online at www.mdpi.com/xxx/s1, Figures S2-S19: 1H- and 13C-NMR spectra of compounds 1-9; Figure S20: Scavenger activity of capsaicin through the DPPH assay using EPR spectroscopy”

11.          In Supplementary Materials, Figure 2, the colours do not match the concentrations and what are C1 C4, is it a compound? It would be better if they were in order C1-C4-C5…and not randomly.

Author Response

The Authors thank the Reviewer for suggestions to improve the quality of the manuscript.

  • Major comments:
  • Compound 6 was not selected for subsequent biological assays due to the low R value (only 4%) in the DPPH assay and the high EC50 value in the TRPV1 functional assay (20.5 µM, low potency). As suggested, the explanation was introduced in the text (lines 165-166).
  • In all the experiments (TRPV1 functional assays, DPPH assay and cell assays) capsaicin, the main natural agonist of the TRPV1 receptor and compound with a known antioxidant action, was used as positive control.
  • Specific comments:
  • In Table 1 the symbol “%” after TRPV1 Efficacy was added.
  • Text from line 136-148 has been transferred to the “Materials and Methods” section, in paragraph 4.2, lines 407-417.
  • In the revised manuscript, lines 143-148.
  • See line 176.
  • We agree with the reviewer and, in fact, in the text (lines 174-176) the Authors underline that only capsaicin and compound 9 are able to increase cell viability. In agreement, the Authors have reformulated the legend of figure 3.
  • Although capsaicin did not exert significant effects on MMP at any concentration used, it is able to protect cells form death. This could be explained taking into consideration that capsaicin is able to inhibit hypoxia-induced HIF-1α accumulation and to increase intracellular oxygen levels (Tae-Hee Han, M. K. Park, H. Nakamura, H. S. Ban. Capsaicin inhibits HIF-1α accumulation through suppression of mitochondrial respiration in lung cancer cells, Biomedicine & Pharmacotherapy, 146, 112500, 2022). This effect could be responsible of the unaltered viability of HaCat cells in hypoxia condition in the presence of capsaicin. In addition to capsaicin only Compound 9 maintains cell viability in hypoxia condition, maybe with a different mechanism of action. Compd 9, in fact, possesses a strong antioxidant activity and is able to counteract ROS formation, thus maintaining also the MMP during hypoxic condition.
  • In the revised manuscript, line 220.
  • TRPV1 activity structure relationships have been discussed (lines 213 to 233) and in fact, the efficacy parameter refers exclusively to the functional assay on TRPV1 receptors. Compound 8 like compounds 1, 2 and 4 have high efficacy (> 65%); however, to continue in the studies, the potency and chemical structure have been also considered. Anyway, the sentence from lines 220 to 221 has been reformulated.
  • In the revised manuscript, line 252.
  • The sentence “Figure S21: Cell viability of capsaicin (caps) and tested lipoic amides 1, 4-5, 7 and 9” has been added in description “Supplementary Materials”, lines 489-490.
  • In Supplementary Materials, Figure 2 has been modified as suggested.

Reviewer 2 Report

This work synthesized nine lipoyl amides in good yields and investigated their biological activities as TRPV1 agonists. The obtained results can provide a good base for the development of new drug or multifunctional food ingredients. Several suggestions are given as follows:

1. The abstract section should include some important results.

2. The abbreviations should be checked carefully. When it was used for the first time, the full name should be given.

3. Short sentence is encouraged to be used in the research paper.

Author Response

The Authors thank the Reviewer for suggestions to improve the quality of the manuscript.

  1. The abstract has been expanded while remaining compliant with the guidelines of the journal. See lines 31-34.
  2. As suggested, the abbreviations were carefully checked throughout the manuscript.
  3. Some sentences have been shortened as required; see lines 55-58, 78-80, 86-90, 238-240.

Round 2

Reviewer 1 Report

The authors have revised everything according to comments. The manuscript can be accepted in present form.